# Pre-Clinical Models of Penetrating Brain Injury: Study Protocol for a Scoping Review

**DOI:** 10.3390/neurosci6020037

**Published:** 2025-04-30

**Authors:** Cindy K. Wong, Jennifer E. Dinalo, Patrick D. Lyden, Gene Sung, Roy A. Poblete

**Affiliations:** 1Department of Neurology, Keck School of Medicine, University of Southern California, Los Angeles, CA 90033, USA; plyden@usc.edu; 2Norris Medical Library, University of Southern California, Los Angeles, CA 90033, USA; dinalo@usc.edu; 3Department of Physiology and Neuroscience, Zilkha Neurogenetic Institute, Keck School of Medicine, Los Angeles, CA 90033, USA; 4Department of Neurology, Los Angeles General Medical Center, Los Angeles County Department of Health Services, Los Angeles, CA 90033, USA; gsung@usc.edu (G.S.); rpoblete@usc.edu (R.A.P.)

**Keywords:** penetrating brain injuries, traumatic brain injuries, pre-clinical model, animal and computational models

## Abstract

Penetrating brain injuries (PBI) constitute a significant subset of traumatic brain injuries, characterized by high morbidity and mortality due to their unique pathophysiological mechanisms. Despite its clinical prevalence in civilian and military settings, progress in translational research remains limited due to a lack of well-characterized pre-clinical models that accurately replicate human PBI. Existing models often fail to adequately simulate critical aspects such as ballistic dynamics, tissue cavitation, and secondary injury cascades, limiting their translational relevance and hindering therapeutic advancements. This scoping review aims to systematically evaluate existing pre-clinical models, including animal, computational, ballistic, and hybrid simulations, to assess their methodological rigor, translational applicability and reported outcome measures. Using PRISMA-ScR guidelines, we will conduct a comprehensive literature search across multiple databases, extracting data on model characteristics, injury induction techniques, histopathological findings, biomolecular markers, and functional assessments. Additionally, bibliometric analyses will provide insights into research trends and gaps in PBI modeling, particularly concerning replicating real-world injury mechanisms and long-term functional outcomes. Through this evaluation, we aim to identify optimal experimental frameworks for studying PBI pathophysiology and recovery mechanisms while informing future model development for therapeutic advancements. The findings from this review will serve as a foundation for advancing pre-clinical PBI research, guiding future model development and therapeutic innovations, and ultimately enhancing treatment strategies and patient outcomes.

## 1. Introduction

Penetrating brain injuries (PBI) represent a unique etiologic subset of traumatic brain injuries (TBI), contributing to over 10% of all TBI cases [1], and are associated with higher mortality rates and more severe clinical outcomes compared to blunt-force TBI [2]. PBI can be broadly categorized based on injury mechanisms, including high-velocity versus low-velocity injuries and blunt–penetrating trauma. High-velocity PBIs are typically caused by projectiles such as bullets or shrapnel, which breach the skull and transfer high kinetic energy to the brain parenchyma, resulting in both a permanent cavity along the projectile path and a temporary cavity caused by shockwave-induced cavitation [3]. In contrast, low-velocity PBIs, such as stab wounds, generate localized damage with minimal cavitation effects. Blunt–penetrating injuries, often caused by impalement or foreign objects, combine direct tissue disruption with compressive forces [3].

Unlike blunt TBI, which primarily involves diffuse axonal injury and coup–contrecoup mechanisms [4], PBI involves complex biomechanical phenomena such as cavitation, shearing forces, and kinetic energy dissipation. The temporary cavity created by high-velocity projectiles expands and collapses rapidly, producing shearing forces that damage neuronal and vascular structures, resulting in unique neuropathological sequelae that differ markedly from those seen in closed-head injuries [3]. Mortality rates for PBI vary by context: civilian settings report an estimated rate of 43.8% [1], whereas military settings show a lower rate of around 18.0% [5], likely due to the availability of immediate, advanced medical and surgical care. Notably, military patients with isolated gunshot wounds to the head are more likely to receive aggressive surgical interventions, such as craniotomy or craniectomy, and experience a lower mortality rate of 27% compared to 38% among their civilian counterparts [6]. PBI remains prevalent in both civilian and combat populations, with PBI frequently leading to significant functional impairment or death.

Despite the prevalence and the severe consequences associated with PBI, there is a substantial gap in translational research regarding pre-clinical models that can accurately replicate the biomechanical and physiological characteristics of this injury observed in human cases [7]. The lack of well-characterized reproducible pre-clinical models impedes our understanding of the pathophysiology and progression of PBI and delays the development of effective therapeutic interventions to mitigate these specific injuries’ primary and secondary damages. Potential experimental models must capture the intricate damage patterns and secondary injury processes of PBI, accounting for complex biomechanical factors such as high-velocity ballistic forces and cavitation. Meaningful biomarkers of disease and outcome should also be identified in the pre-clinical space. In the current limited literature, wide variability in experimental models further limits results in comparisons [8]. Consequently, an urgent need exists to evaluate and enhance pre-clinical models that can accurately mimic the human condition.

Here, we have developed a protocol for a scoping review that aims to provide a replicable framework to comprehensively evaluate existing pre-clinical models of PBI in the literature, with specific goals to assess pre-clinical methodological vigor, validity, reliability, and translational relevance in PBI research. Through this evaluation, we aim to identify and prioritize gaps in the research landscape and provide guidance for developing robust experimental frameworks to understand disease pathophysiology, recovery mechanisms, potential therapeutic targets, and prognostication after PBI, ultimately improving functional recovery and survival outcomes for patients with PBI.

## 2. Methods and Analysis

### 2.1. Study Design and Protocol Registration

This protocol follows the PRISMA-ScR (Preferred Reporting Items for Systematic Reviews and Meta-Analyses Extension for Scoping Reviews) guidelines to ensure transparency and reproducibility [9]. The review is registered with the Open Science Framework (OSF) to prevent duplication of efforts and maintain methodological transparency.

### 2.2. Study Selection and Eligibility Criteria

Study selection for this scoping review will adhere to specific eligibility criteria. Only original, peer-reviewed research articles focusing on experimental pre-clinical models of PBI will be included. These studies may utilize animal, computational, ballistic, or hybrid models combining these approaches. This review excludes conference abstracts, review articles, clinical trials lacking pre-clinical methods, and meta-analyses.

Studies employing any animal species as PBI models, including rodents (e.g., rats and mice), rabbits, swine, canines, ferrets, and non-human primates, will be considered. Computational models that simulate the biomechanics of PBI are also eligible. However, studies that exclusively describe blunt-force TBI models or other head injuries without penetrating mechanisms will be excluded.

Interventions in this review encompass any form of PBI induction, such as ballistic injuries, controlled cortical impact with penetration, or stab-like injuries. The specific method of injury induction must be clearly described, and studies must provide adequate methodological details to be experimentally reproducible to be included.

The primary aim of this scoping review will be to extract, describe, and synthesize methodologies of mimicking and studying pre-clinical PBI. Similarities and differences between models will be described, specifically comparing animal and non-animal models and, within animal models, comparing large and small animal models. This comparative analysis will evaluate each model type’s rigor, utility, translational relevance, and methodological variability. Secondary aims include a descriptive analysis of outcome measures used in PBI models, including histopathological findings (e.g., pathology, staining, gross lesion size estimation), diagnostic imaging results (e.g., magnetic resonance imaging), in vivo biomolecular markers (e.g., cytokine levels, metabolomics, gene expression), and behavioral assessments (e.g., neurological scoring, motor, and cognitive function tests) in animal models, as well as outcome measures used in non-animal models. Studies that lack specific outcome data to validate or justify the PBI model will be excluded from this review. An additional secondary aim will be the bibliometric analysis of publications meeting the eligibility criteria of this protocol, allowing serial longitudinal assessment of research interest and productivity in the field of PBI.

Finally, this scoping review will consider articles in both English and non-English languages. Non-English papers will be translated using available resources, such as professional translation services or bilingual researchers, to ensure accurate interpretation. There will be no publication date restrictions, and studies from the inception of each database will be included.

### 2.3. Study Organization

EndNote X8.2^TM^ will be used as a citation management tool to organize and manage references. Covidence, a continuously updated web-based platform, will assist in importing and de-duplicating search results from various databases, facilitating the screening and selection process. Covidence will also store and manage full-text articles and ensure accurate and consistent screening during full-text review.

### 2.4. Search Strategy

The search strategy for this scoping review was developed based on a validated search approach for pre-clinical studies from the University of Southern California, Norris Medical Library (Table 1). We will comprehensively search the peer-reviewed literature across three electronic databases: PubMed/MEDLINE, Embase, and Web of Science. The search will be unrestricted by date or language; however, we will exclude human studies without pre-clinical modeling, conference abstracts, reviews, and meta-analyses to focus on the relevant pre-clinical literature. These databases will be searched using a combination of controlled vocabulary (e.g., Medical Subject Heading terms) and free-text keywords related to PBI and pre-clinical models.

### 2.5. Reference List

Reference lists of systematic and scoping reviews identified from databases will be manually reviewed for additional relevant studies.

### 2.6. Study Selection Process

The study selection process for this scoping review will be conducted in two stages. First, a title and abstract screening will be performed by two independent reviewers, who will assess all retrieved studies for relevance according to the eligibility criteria. Each study will be categorized as “include”, “exclude”, or “uncertain”. In the second stage, the full texts of studies deemed “include” or “uncertain” will undergo review. Any discrepancies in study selection will be resolved through discussion and consultation with a third reviewer. The selection process will be thoroughly documented in a PRISMA flow diagram [10] detailing the reasons for exclusion at each stage.

### 2.7. Data Extraction

Data extraction will prioritize the primary aim of identifying and analyzing the methodology of PBI modeling, including the types of models used, such as animal or computational, and the specific techniques applied to simulate PBI. Collected information will include primarily study characteristics, such as author(s), year of publication, country of origin, and funding sources. Model characteristics will consist of the type of model used (animal, computational, or hybrid) and details like species, strain, sex, weight, and the number of animals involved. Information on injury induction will detail the method used to induce PBI, including device specifications, injury parameters (e.g., velocity, trajectory, depth), and anesthesia protocols. Secondary aims data will also be collected, including histopathology, radiographic imaging findings, behavioral assessment, in vivo biomolecular markers in animal models, and any measures used in non-animal models of PBI.

Additionally, bibliometric data, such as citation counts, journal of publication, year of publication, institutions, author groups, and collaborative networks, will be gathered. Two reviewers will perform data extraction independently, and discrepancies will be resolved by consensus. If consensus cannot be reached, a third reviewer will be consulted. A pilot test will be conducted on a subset of studies to refine the data extraction process.

### 2.8. Quality Assessment

The included studies’ methodological quality and risk of bias will be assessed using SYRCLE’s risk of bias tool for animal studies and a customized checklist for computational models that will include specific criteria such as model validation, sensitivity analysis, and transparency of code [11]. The assessment will focus on selection, performance, detection, attrition, and reporting biases, while also evaluating critical aspects of computational models such as reproducibility, accuracy, and robustness. Other potential biases will also be evaluated, such as study sponsorship or outcome reporting bias. Two reviewers will independently assess each study’s quality, and consensus will resolve disagreements, as previously discussed.

### 2.9. Data Synthesis

Given the anticipated heterogeneity in study designs, injury models, and outcome measures, a narrative synthesis will be employed to summarize the findings. Following the structured approach outlined in the Guidance on the Conduct of Narrative Synthesis, this process will involve developing a theoretical framework to contextualize the results, conducting a preliminary synthesis to organize and describe the study data, and exploring relationships to explain differences across studies [12]. The synthesis will categorize and compare methodologies used in various pre-clinical PBI models, highlighting similarities, differences, and their relevance to human PBI. Secondary outcomes, such as bibliometric trends, computational analyses, histopathological findings, and clinical biomarkers, will also be summarized to evaluate the models’ effectiveness and reproducibility. This approach will assess the robustness of the synthesis by evaluating study quality and potential biases, ultimately identifying research gaps and suggesting future directions to inform better modeling practices.

### 2.10. Assessment of Bias and Confidence in Evidence

The risk of bias across the included studies will be evaluated, and publication bias will be assessed using funnel plots, if applicable. The confidence in the cumulative evidence will be determined using the GRADE (Grading of Recommendations Assessment, Development, and Evaluation) approach [13]. This validated tool will classify the certainty and quality of evidence into four levels: very low, low, moderate, or high. Narrative recommendations will be provided, particularly emphasizing the clinical implications of the study results, ensuring that these recommendations are grounded in the rigorously assessed quality of evidence.

## 3. Discussion

PBI is a complex and phenotypically unique condition that remains significantly understudied in clinical and pre-clinical translational research. The latter is critical for developing therapeutic strategies with human applications. One of the most urgent needs in PBI research is the development of feasible and reproducible pre-clinical models that can accurately mimic the high-force impacts, ballistic trajectories, and other biomechanical characteristics that define human PBI.

Certain limitations in replicating the full spectrum of injury mechanisms and outcomes in human PBI using animal models should be discussed. Smaller mammals, particularly rodents, are commonly employed in pre-clinical research due to their cost-effectiveness and ease of handling. Nevertheless, the anatomical and physiological disparities between these animals and humans—including differences in brain size, structural composition, and healing processes—raise significant concerns regarding their capacity to accurately simulate the initial injury patterns and progression of diseases observed in human PBI. Rodent models, for instance, possess inherent constraints stemming from their lissencephalic brain structure, which markedly diverges from the gyrencephalic brains characteristic of humans [14]. This variation affects how applicable the findings are, potentially limiting the application of research outcomes to clinical practice.

Conversely, larger animal models, such as swine and non-human primates, may present a more appropriate approach for PBI research. These larger species closely resemble human neuroanatomy and injury responses, offering more accurate modeling advantages [15]. Swine, in particular, have been utilized in PBI studies owing to their gyrencephalic brain configuration and comparable ratios of white to gray matter, which provide a more precise representation of human brain structure [16]. Additionally, larger animal models are better suited for simulating the high-velocity, high-impact injuries that typify penetrating brain trauma [17]. While employing larger animals in research provides considerable advantages, it can pose significant challenges. These challenges encompass ethical considerations, such as the need to ensure the welfare of the animals and the potential for public concern over the use of larger animals in research, elevated costs, and complexities of managing such animals within laboratory environments, including the need for specialized facilities and trained personnel. Although larger animal models may afford a more authentic representation of human brain injury patterns, these practical limitations can restrict their widespread application in research initiatives.

In addition to improving animal models, non-animal models, including computational and biomechanical models, can simulate injury mechanics and kinetics without animal studies’ ethical and logistical constraints. These models, such as Finite Element Models (FEM) that simulate stress distribution and tissue deformation during projectile impacts [18] and computational biomechanics models that examine how skull–brain interface modeling influences injury predictions [19] provide valuable insights into how different forces angles and velocities impact the brain. Additionally, platforms such as The Virtual Brain (TVB) offer simulations of brain network dynamics, enabling the exploration of the functional consequences of injuries [20]. These models provide insights into biological responses in PBI, assist in predicting injury outcomes, and contribute to optimizing intervention strategies. However, the reliability of these computational models depends on the quality of the input data and the extent of their validation against empirical findings. The limited availability of robust, well-validated computational models highlights the need for ongoing innovation and investment in this approach. The advancement of non-animal models can complement traditional animal studies and reduce reliance on live subjects while gaining valuable insights into the mechanics and pathology of PBI.

In vitro models have also emerged as indispensable tools for investigating PBI, offering controlled environments to examine cellular responses to injury. Recent advancements in three-dimensional (3D) culture techniques have substantially improved the physiological relevance of these models, successfully replicating brain tissue’s complex structure and mechanical properties. For example, a 3D in vitro neuronal compression model has been established to assess the effects of impact strain and strain rate on neuronal viability and morphology [21]. This model elucidates the mechanical thresholds that are responsible for neuronal damage. Additionally, a 3D tissue model developed for traumatic brain injury research has successfully demonstrated excitotoxicity—a critical element of secondary injury mechanisms—by illustrating increased glutamate release in conjunction with a pattern of cell death that parallels in vivo observations [22]. These in vitro models facilitate high-throughput screenings of potential therapeutic agents and allow for precise experimental manipulations, addressing challenges typically encountered in live animal research. However, it is essential to recognize the limitations inherent in in vitro models, particularly their inability to incorporate systemic factors, such as immune responses and blood flow, and fully replicate the in vivo environment. Therefore, it is imperative to utilize these models in conjunction with animal and computational studies to understand PBI pathophysiology comprehensively.

Selecting appropriate outcome measures is vital in modeling PBI, as both traditional methods (e.g., histopathological analyses and imaging techniques) and functional assessments together provide a comprehensive understanding of injury effects. Recent advancements in neuroimaging, such as diffusion tensor imaging and functional magnetic resonance imaging, have enhanced the ability to visualize structural and functional alterations following PBI, allowing for the assessment of axonal integrity and brain connectivity and offering a more detailed understanding of injury mechanisms and potential recovery patterns [23]. Additionally, molecular and cellular outcome measures, including inflammation, oxidative stress, and cell death biomarkers, have emerged as valuable tools to complement traditional histological analyses, further enriching the characterization of injury progression [24]. Functional assessments are needed to translate preclinical findings into clinical contexts. Novel approaches, such as automated gait analysis and rodent cognitive task paradigms (e.g., Morris Water Maze), are increasingly utilized to quantify functional impairments with greater precision, marking significant progress in the field [25]. However, it is crucial to consider species-specific behaviors and responses when translating findings from animal models to human conditions, as rodents, while displaying robust spatial navigation capabilities, may not directly correlate with human cognitive functions, necessitating cautious interpretation when extrapolating outcomes to clinical scenarios. Integrating multimodal approaches that combine behavioral, imaging, and molecular analyses may enhance the accuracy and relevance of outcome measures, providing a holistic understanding of PBI pathophysiology while aiding in evaluating the efficacy of therapeutic interventions through diverse endpoints.

Lastly, therapeutic interventions should also be explored and rigorously tested using advanced pre-clinical models. For example, one promising approach is the application of moderate hypothermia, which has been shown to reduce secondary injury processes, including inflammation, apoptosis, and excitotoxicity [26]. Implementing hypothermia in experimental settings can help determine its potential to improve neurological outcomes, reduce mortality, and enhance recovery following PBI. Given its success in other types of traumatic brain injuries, moderate hypothermia warrants investigation as a targeted therapeutic strategy for PBI, guided by robust experimental and computational models [27].

As interest in PBI research grows, there is a need to establish validated pre-clinical models that enhance our understanding of disease pathophysiology and recovery mechanisms. A well-constructed pre-clinical model is a foundational tool for evaluating potential pharmacological therapies, surgical interventions, and rehabilitation strategies, all aimed at improving clinical outcomes for civilian and military populations affected by PBI. This scoping review protocol delineates a comprehensive framework for evaluating existing pre-clinical PBI models, primarily focusing on their methodologies and outcome measures. By systematically analyzing the strengths, limitations, and translational potential of various models, the insights gained will prove invaluable to researchers and clinicians in this developing field. This initiative underscores the importance of interdisciplinary collaboration in establishing a benchmark for future advancements in translational research pertaining to PBI.

### Limitations

This scoping review is limited to pre-clinical models of penetrating brain injuries. It does not encompass clinical research, which would require a separate review to assess the validity and generalizability of the study to clinical settings. A notable constraint in the current literature is the limited development and availability of non-animal models, potentially restricting the generalizability of our findings to alternative systems. Additionally, there is a risk of publication bias, as studies with positive or noteworthy results are more likely to be published. At the same time, those with negative or inconclusive findings may be underrepresented. While this review aims to evaluate the precision and accuracy of the included models, it is essential to recognize that full validation of these models based solely on published data is not feasible. Moreover, relevant studies may be missed if they are published in non-English languages, indexed in databases outside our search scope, or only available as abstracts, potentially impacting the scope of this review. These limitations should be considered when interpreting the findings of this review.

## 4. Conclusions

Several key recommendations can guide the future of PBI modeling research. First, there is a need for developing standardized and validated models that can be consistently replicated across laboratories. These models should be robust enough to accurately capture the biomechanical and physiological characteristics of PBI, ensuring reproducibility and reliability. Enhancing computational and biomechanical models will also enable more precise simulations of injury dynamics and functional impairments. Integrating these computational models with real-world clinical data will further validate their accuracy and enhance their translational relevance.

To provide comprehensive insights into injury mechanisms and therapeutic responses, the use of hybrid models that combine animal experimentation with computational simulations should be prioritized. By bridging traditional and advanced modeling approaches, researchers can achieve a more holistic understanding of PBI pathophysiology. Emphasis should also be placed on data transparency and sharing through open-access platforms, fostering collaboration and reproducibility.

Lastly, interdisciplinary collaboration between bioengineers, neuroscientists, and clinicians will be crucial to advancing PBI research. Investment in advanced technologies, including machine learning and data-driven approaches, can help identify predictive biomarkers and refine model parameters, ultimately guiding therapeutic innovation. Ensuring that developed models align with clinical relevance and practical applicability will enhance their potential to inform treatment strategies and improve patient outcomes.

By incorporating these recommendations, future research can overcome current limitations, fostering the development of robust, reliable, and translatable PBI models that better serve the needs of research and clinical practice.

## Figures and Tables

**Table 1 neurosci-06-00037-t001:** Database Search Strategy.

Database Search Strategy
**Medline (PubMed) Search Strategy**
(“head injuries, penetrating”[MeSH Terms] OR “penetrat* head injur*” OR “penetrat* traumatic head injur*” OR “penetrating head trauma*” OR “penetrat* brain injur*” OR “penetrat* traumatic brain injur*” OR “penetrating brain trauma*” OR “penetrating craniocerebral trauma*” OR “penetrat* head wound” OR “penetrat* traumatic head wound” OR “penetrating head wound” OR “penetrat* brain wound” OR “penetrat* traumatic brain wound” OR “penetrating brain wound” OR “penetrating craniocerebral wound” OR “penetrat* head wounds” OR “penetrat* traumatic head wounds” OR “penetrating head wounds” OR “penetrat* brain wounds” OR “penetrat* traumatic brain wounds” OR “penetrating brain wounds” OR “penetrating craniocerebral wounds” OR “penetrat* skull injur*” OR “penetrat* traumatic skull injur*” OR “penetrating skull trauma*” OR “penetrat* skull wound” OR “penetrat* traumatic skull wound” OR “penetrating skull wound” OR “penetrat* skull wounds” OR “penetrat* traumatic skull wounds” OR “penetrating skull wounds”) AND (“Animals”[MeSH] OR “rodent*” OR “mouse*” OR “mice” OR “rabbit*” OR “swine” OR “porcine” OR “canine” OR “ferret” OR “primate” OR “Models, Theoretical”[Mesh] OR “animal model” OR “Theoretical model” OR “animal models” OR “Theoretical models” OR “computational” OR “biomechanic*” OR “simulat*” OR “ballistic” OR “bibliometric” or “bibliographic” OR “predict*”)
**Embase Search Strategy**
(‘penetrating head injury’/exp OR ‘penetrat* head injur*’ OR ‘penetrat* traumatic head injur*’ OR ‘penetrating head trauma*’ OR ‘penetrat* brain injur*’ OR ‘penetrat* traumatic brain injur*’ OR ‘penetrating brain trauma*’ OR ‘penetrating craniocerebral trauma*’ OR ‘penetrat* head wound’ OR ‘penetrat* traumatic head wound’ OR ‘penetrating head wound’ OR ‘penetrat* brain wound’ OR ‘penetrat* traumatic brain wound’ OR ‘penetrating brain wound’ OR ‘penetrating craniocerebral wound’ OR ‘penetrat* head wounds’ OR ‘penetrat* traumatic head wounds’ OR ‘penetrating head wounds’ OR ‘penetrat* brain wounds’ OR ‘penetrat* traumatic brain wounds’ OR ‘penetrating brain wounds’ OR ‘penetrating craniocerebral wounds’ OR ‘penetrat* skull injur*’ OR ‘penetrat* traumatic skull injur*’ OR ‘penetrating skull trauma*’ OR ‘penetrat* skull wound’ OR ‘penetrat* traumatic skull wound’ OR ‘penetrating skull wound’ OR ‘penetrat* skull wounds’ OR ‘penetrat* traumatic skull wounds’ OR ‘penetrating skull wounds’) AND (‘animals’/exp OR ‘rodent*’ OR ‘mouse*’ OR ‘mice’ OR ‘rabbit*’ OR ‘swine’ OR ‘porcine’ OR ‘canine’ OR ‘ferret’ OR ‘primate’ OR ‘theoretical model’/exp OR ‘animal model’ OR ‘Theoretical model’ OR ‘animal models’ OR ‘Theoretical models’ OR ‘computational’ OR ‘biomechanic*’ OR ‘simulat*’ OR ‘ballistic’ OR ‘bibliometric’ or ‘bibliographic’ OR ‘predict*’)
**Web of Science Search Strategy**
(“penetrat* head injur*” OR “penetrat* traumatic head injur*” OR “penetrating head trauma*” OR “penetrat* brain injur*” OR “penetrat* traumatic brain injur*” OR “penetrating brain trauma*” OR “penetrating craniocerebral trauma*” OR “penetrat* head wound” OR “penetrat* traumatic head wound” OR “penetrating head wound” OR “penetrat* brain wound” OR “penetrat* traumatic brain wound” OR “penetrating brain wound” OR “penetrating craniocerebral wound” OR “penetrat* head wounds” OR “penetrat* traumatic head wounds” OR “penetrating head wounds” OR “penetrat* brain wounds” OR “penetrat* traumatic brain wounds” OR “penetrating brain wounds” OR “penetrating craniocerebral wounds” OR “penetrat* skull injur*” OR “penetrat* traumatic skull injur*” OR “penetrating skull trauma*” OR “penetrat* skull wound” OR “penetrat* traumatic skull wound” OR “penetrating skull wound” OR “penetrat* skull wounds” OR “penetrat* traumatic skull wounds” OR “penetrating skull wounds”) AND (“rodent*” OR “mouse*” OR “mice” OR “rabbit*” OR “swine” OR “porcine” OR “canine” OR “ferret” OR “primate” OR “animal model” OR “Theoretical model” OR “animal models” OR “Theoretical models” OR “computational” OR “biomechanic*” OR “simulat*” OR “ballistic” OR “bibliometric” or “bibliographic” OR “predict*”)

## Data Availability

Data are contained within the article.

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
