# Peer review of "Pre-Clinical Models of Penetrating Brain Injury: Study Protocol for a Scoping Review"

_neurosci, 2025, doi:10.3390/neurosci6020037_

Round 1
Reviewer 1 Report
Comments and Suggestions for Authors
The article presents a well-structured protocol for a scoping review of pre-clinical penetrating brain injury (PBI) models. The rationale for the review is clearly articulated, highlighting the need for better translational models to improve treatment strategies for this severe type of traumatic brain injury (TBI). The use of PRISMA-ScR guidelines demonstrates a commitment to transparency and methodological rigor. However, there are some points and suggestions that should be addressed to further improve the clarity and professionalism of the current review.
- Abstract
- The abstract mentions "animal and computational simulations," the authors should consider adding a word or two to indicate other model types that might be included (e.g., "ballistic" or "hybrid" models) as mentioned later in the text.
- The authors should mention the specific gaps in current PBI modeling to strengthen the impact that the review aims to address.
- Be consistent with hyphenation all over the manuscript (e.g., "pre-clinical" vs "preclinical"). Also check the consistency of introducing and using abbreviations.
- Introduction
- Consider briefly mentioning or categorizing different subtypes of PBI (e.g., low-velocity vs. high-velocity, blunt-penetrating, etc.) in the introduction before discussing their influence on the pathophysiology or the relevance of specific animal or computational models.
- The introduction should contain a slightly stronger statement of the review's specific aims, such as what aspects of the models will be assessed (e.g., validity, reliability, translational relevance).
- Methods and Analysis
- In Table 2 (Study Selection Eligibility Criteria): Consider adding a row to state the rationale for including studies published in languages other than English. As the following:
|
Criteria |
Inclusion |
Exclusion |
|
Language |
Both English and non-English publications. |
[None specified] |
|
Rationale (New) |
Including non-English publications helps to minimize publication bias and ensures a more comprehensive review of the available literature on pre-clinical PBI models. Translation services will be utilized as needed. |
|
- The authors should provide more detail about the "custom checklist" for computational models. What specific criteria will be used to assess the quality and validity of these models? (e.g., model validation, sensitivity analysis, transparency of code).
- Discussion:
- The sentence from line 210 to 214 is too long. Consider breaking it into smaller lines.
- Mention some examples of the computational and biomechanical models of PBI with references.
- Consider adding a separate section of future recommendations with specific points.
Author Response
Comment 1: Abstract
Response: Animal, computational, ballistic, and hybrid simulations have been added to reflect mentioned later text. Specific gaps mentioned particularly concerning the replication of real-world injury mechanisms and long-term functional outcomes added. Hyphenation all consistent now with pre-clinical. Abbreviations revised to PBI throughout article.
Comment 2: Introduction
Response: Mentioned different subtypes of PBI including low-velocity vs. high-velocity, blunt-penetrating. Introduction included stronger statement of aspects of models assessed including validity, reliability, translational relevance.
Comment 3: Methods and Analysis
Response: Included rationale within the text line for languages in the table. Custom checklist for computational models more detailed with specific criteria used to assess the quality and validity highlighted.
Comment 4: Discussion
Response: Divided sentence of 210-214 into short concise sentences. Included examples of computations and biomechanical models with references. Added a conclusion with future recommendations.
Reviewer 2 Report
Comments and Suggestions for Authors
The utility of PBI models needs to be considered in more detail.
The potential for therapeutic intervention needs to be considered, including moderate hypothermia.
Th predictive validity of PBI models also needs to be considered
Author Response
Comment 1:
Utility of PBI models highlighted in more detail in introduction and conclusion.
Comment 2:
Future direction of therapeutic intervention included in the conclusion, including hypothermia
Comment 3:
Custom checklist for computational models more detailed with specific criteria used to assess the quality and validity highlighted.
Reviewer 3 Report
Comments and Suggestions for Authors
Dear Authors,
Thank you for submitting your manuscript entitled "Pre-Clinical Models of Penetrating Brain Injury: Study Protocol for a Scoping Review", to be considered for publication in the journal "NeuroSci".
After having read your paper carefully, I have to make the following comments:
-There is an extensive section covering the "Methods", which is not necessary, at least in such detail.
-On the other hand, a section regarding the "Results" of the study is totally missing.
-The "Discussion" section is too short and discuss only generalities but not specific things, related to the current study. This is evident from the fact that only two references are included here.
Kind regards,
The Reviewer
Author Response
Dear Reviewer,
Thank you for your thoughtful feedback and for taking the time to carefully read our paper. We greatly appreciate your comments and suggestions.
Regarding the "Methods" section, we would like to clarify that the detailed presentation was intentional, as this paper was designed as a study protocol. The purpose of providing a comprehensive methodology was to ensure that the study design is transparent, reproducible, and clearly articulated. However, we understand your concern and would greatly appreciate any specific suggestions on which aspects of the methods could be shortened without compromising clarity or completeness.
Concerning the "Results" section, we would like to note that as this is a study protocol, no results are yet available, as the study has not been performed. We do include a data analysis section to outline how the data will be processed and interpreted once collected. If you have specific recommendations on what aspects or preliminary insights could be added to this section, we would be glad to consider them.
Additionally, we acknowledge your feedback regarding the "Discussion" section. Based on your suggestion, we have made efforts to extend the discussion to include more specific insights related to the current study rather than just generalities. Furthermore, we have incorporated additional references to support and substantiate the discussion. We have also added a conclusion section to better summarize the study’s potential impact and future directions.
We are grateful for your valuable input, as it has helped us to enhance the quality and completeness of the manuscript. Please let us know if you have any further suggestions or recommendations.
Kind regards,
Cindy Wong DO
Round 2
Reviewer 3 Report
Comments and Suggestions for Authors
Dear Authors,
Thank you for your prompt response.
Regarding the "Methods" section, as I mentioned before, it is very detailed and can be quite boring for the reader to read. You mention everything twice, once in the Table and once again within the text. You should shorten it in such a way that it will be more concise but with the appropriate amount of information.
Even though you have somehow tried to improve the "Discussion" section, these amendments are not sufficient enough; I have been expecting to see more regarding my comment/recommendation on this.
Only in the "Conclusions" section you have made considerable modifications.
Kind regards,
The Reviewer
Author Response
Dear Reviewer,
Thank you for your valuable feedback and suggestions.
We have carefully addressed your comments regarding the "Methods" section. The tables previously included in this section have been removed, and the text has been edited to be more concise while retaining the necessary amount of information.
Additionally, we have extended the "Discussion" section to include further details and specifics about the pre-clinical models investigated, as per your recommendation. We hope that these revisions meet your expectations and enhance the clarity and quality of the manuscript.
Thank you once again for your thoughtful input and guidance.
Sincerely,
Cindy Wong, DO
Round 3
Reviewer 3 Report
Comments and Suggestions for Authors
Dear Authors,
Thank you for sharing the revised version of your manuscript.
The only comment that I have to make, is to exclude the paragraph with the two new citations "Therapeutic interventions should also be explored and rigorously tested using advanced pre-clinical models... guided by robust experimental and computational models [27]." (Lines 334-341) from the Conclusion section. This should be placed in the Discussion section.
Good luck with the publication of your paper.
Kind regards,
The Reviewer
Author Response
Dear Reviewer,
Thank you for your valuable feedback and for taking the time to review our revised manuscript.
As per your suggestion, we have moved the paragraph containing the two new citations (Lines 334–341) from the Conclusion section to the Discussion section.
We appreciate your guidance and support.
Kind regards,
Cindy Wong, DO